# Non-Invasive Analysis of Actinic Keratosis before and after Topical Treatment Using a Cold Stimulation and Near-Infrared Spectroscopy

**DOI:** 10.3390/medicina56090482

**Published:** 2020-09-21

**Authors:** Silvia Seoni, Paola Savoia, Federica Veronese, Elisa Zavattaro, Vanessa Tarantino, Kristen M. Meiburger

**Affiliations:** 1PoliToBIOMed Lab, Biolab, Department of Electronics and Telecommunications, Politecnico di Torino, Corso Duca degli Abruzzi 24, 10129 Torino, Italy; kristen.meiburger@polito.it; 2Dermatology Unit, Department of Health Science, University of Eastern Piedmont, 28100 Novara, Italy; paola.savoia@med.uniupo.it (P.S.); federica.veronese@med.uniupo.it (F.V.); vanessa.tarantino30@gmail.com (V.T.); 3Dermatology Unit, Department of Translational Medicine, University of Eastern Piedmont, 28100 Novara, Italy; elisa.zavattaro@med.uniupo.it

**Keywords:** signal processing, near-infrared spectroscopy, actinic keratosis, field cancerization

## Abstract

*Background and objectives*: The possible evolution of actinic keratoses (AKs) into invasive squamous cell carcinomas (SCC) makes their treatment and monitoring essential. AKs are typically monitored before and after treatment only through a visual analysis, lacking a quantitative measure to determine treatment effectiveness. Near-infrared spectroscopy (NIRS) is a non-invasive measure of the relative change of oxy-hemoglobin and deoxy-hemoglobin (O_2_Hb and HHb) in tissues. The aim of our study is to determine if a time and frequency analysis of the NIRS signals acquired from the skin lesion before and after a topical treatment can highlight quantitative differences between the AK skin lesion area. *Materials and Methods*: The NIRS signals were acquired from the skin lesions of twenty-two patients, with the same acquisition protocol: baseline signals, application of an ice pack near the lesion, removal of ice pack and acquisition of vascular recovery. We calculated 18 features from the NIRS signals, and we applied multivariate analysis of variance (MANOVA) to compare differences between the NIRS signals acquired before and after the therapy. *Results*: The MANOVA showed that the features computed on the NIRS signals before and after treatment could be considered as two statistically separate groups, after the ice pack removal. *Conclusions*: Overall, the NIRS technique with the cold stimulation may be useful to support non-invasive and quantitative lesion analysis and regression after a treatment. The results provide a baseline from which to further study skin lesions and the effects of various treatments.

## 1. Introduction

Non-melanoma skin cancers (NMSC) are the most common human cancers, particularly in the Caucasian population; in Italy, the prevalence of the actinic keratoses (AKs) is estimated to be around 30% [1], making them a major health problem, particularly considering their possibility of evolution into invasive squamous cell carcinomas (SCC).

In the last few years, the treatment of AK has changed and the concept of “lesion directed therapy” has been overtaken by “field therapy”. It is well-known that chronic exposure to ultraviolet radiation (UVR) is the main risk factor for NMSC and UVR-induced photo damage can cause a field of actinic damage. This field usually presents clinically visible AKs and subclinical cell alterations (histologic atypia) that represent a substrate for new lesion development [2,3,4,5]. The early treatment of all lesions (clinically visible and subclinical) must be considered in order to reduce the possible risk of evolution into squamous cell carcinoma (SCC) [6,7,8,9,10,11].

AKs are diagnosed by the dermatologist through a clinical exam and the support of portable dermatoscopes, but often this method is not sufficient to ensure that the treatment has been effective or to discriminate between AK and SCC. Non-invasive imaging techniques, such as in vivo confocal laser microscopy [12] and optical coherence tomography (OCT), can provide complementary information [13] but require expensive instrumentation and long scanning times. Clinically, dermatologists currently rely on biopsy or excision with histological examination.

NIR spectroscopy uses low power radiation in a safe region of the electromagnetic spectrum, the near-infrared window, that is not dangerous and does not cause pathological alterations in cells. The NIR radiations interact with endogenous chromophores such as the oxygenated (O_2_Hb) and deoxygenated hemoglobin (HHb), water, and lipids. To evaluate blood flow and blood oxygen saturation, the hemoglobin concentration is monitored through light scattering and tissue absorption. The NIRS signal calculates the light intensity from the near-infrared light source after the light has passed through the analyzed tissue. The hemoglobin concentration is studied through the evaluation of the differences in the absorption spectra of the O_2_Hb and HHb. This technique has several advantages: it has high temporal resolution, it is not invasive or painful, and it is inexpensive and portable. On the other hand, there are some downsides, such as low spatial resolution or depth perception.

The literature reports numerous studies that have investigated the usefulness of near-infrared (NIR) spectroscopy as a supportive tool in the diagnosis and evaluation of skin tumors. When cells undergo transformation, the composition of their structure changes and these changes can be evaluated by using NIRS analysis and skin impedance techniques [14,15,16,17]. Hägerlind et al. investigated the usefulness of near-infrared (NIR) and skin impedance spectroscopy as a supportive tool in the diagnosis and evaluation of skin tumors in primary healthcare [14]. This tool was able to help with the diagnosis of malignant, premalignant and benign skin lesions, as a complement to the visual assessment. Furthermore, the NIR and impedance spectra were able to distinguish between cutaneous malignant melanoma (CMM), dysplastic nevi (DN) and other benign nevi (N) [15]. The data were analyzed with a multivariate technique and showed good results for the discrimination of malignant melanomas (sensitivity and specificity were 83% and 95%, respectively). Another interesting result was the usefulness of a combined probe head for simultaneous, time-saving NIR spectroscopy and skin impedance measurements to diagnose the skin lesions [17]. Shirkavand et al. used NIRS for the in vitro determination of the refractive index of human skin cell lines for melanoma, fibroblast and adipose samples with the purpose of allowing screening of normal and abnormal cells for probable alterations in a non-invasive method [18].

In the literature, data concerning the use of NIRS in AK are still lacking, besides a few studies that have previously investigated its vascularization. Carpenter et al. [19] have recently evaluated various cutaneous lesions through the use of optical spectroscopy that permits the measurement of different parameters, including the HHb and O_2_Hb concentration and saturation. Their study aimed to identify the benign or malignant nature of a cutaneous lesion through such techniques and to establish a concordance between spectroscopy and biopsy. They showed that malignant lesions were hypoxic in comparison with healthy skin, and the low specificity of spectroscopy in recognizing AK is noteworthy.

In addition, our previous study [20] showed the different vascular oxygenation states between healthy skin and AK lesions. Using the same approach (NIRS and cold stimulation), here, we extend upon our previous results about the discrimination between the AK lesions and healthy skin and improve the analysis with the evaluation of the effects of the topical treatment. In particular, we use NIR spectroscopy to analyze skin lesions, without the skin impedance analysis, simplifying data acquisition. The NIR spectroscopy analysis is used as a non-invasive technique to study relative concentration changes in O_2_Hb and HHb in AK arising in the field of actinic damage (also known as Field Cancerization -FC) before and after topical treatment. Cold stimulation was added in order to evaluate the vascular response near the skin lesion. In this way, we reduced the time and cost of the acquisition, obtaining a non-invasive and useful tool that gives us information about the skin lesions and their vascular condition. The cold stimulation is a simple ice pack application near the probes (for 1.5 min), that generates a vasoconstriction (during ice pack application) and subsequent vascular recovery (after ice pack removal). With the cold stimulation, in the same data acquisition session, we can evaluate the NIRS signals in different conditions. The ice pack is placed near the probes, without touching them, in order to minimize noise or artefacts in the signal recording. The stimulus is not painful for the patient, only causing slight discomfort if applied on the head. Furthermore, the ice pack is not costly, it is readily available, and it is easy to apply.

The aim of this study is to verify whether the non-invasive time and frequency NIRS signal analysis technique can provide information about the differences between the AK skin lesion area before and after a topical treatment, providing information to monitor the response to treatment or potential regression and progression of skin lesions.

## 2. Materials and Methods

### 2.1. Study Population

The study was performed at the Dermatologic Outpatient Clinic in Novara (Italy) in collaboration with the Polito^BIO^Med Lab. of Department of Electronics and Telecommunications, Politecnico of Turin (Italy). The study protocol was approved by the Local Ethics Committee and was conducted following the Declaration of Helsinki. All patients signed an informed consent before being included in the study.

We enrolled 22 male subjects with a mean age of 74 ± 9 years, during a period of 11 months, from February 2018 to January 2019. The patients presented FC and clinical evidence of AKs (grade I and II) on the scalp, diagnosed by an expert dermatologist through clinical exam and a portable dermatoscope (HEINE DELTA^®^20 T, Heine Optotechnik, Herrsching, Germany). The patients had not been previously treated with any therapy for FC and/or AKs.

After the diagnosis, the affected skin area was treated with topical Imiquimod 3.75% (IMI), according to the standard treatment modality: cream was applied once a day for 2 weeks, and then during the following 14 days, the application of the cream was stopped; finally, the treatment was repeated for another 2 weeks. Imiquimod is an immunomodulating topical agent (toll-like receptor 7 agonist) that stimulates the innate and adaptative immune pathways and induces cytokine production, like interferon (IFN)-a, tumor necrosis factor (TNF)-a, interleukin (IL)-6, and IL-8. Furthermore, Imiquimod, at higher concentrations, induces apoptosis of tumor cells. Imiquimod has also shown anti-angiogenic properties, which are due to the production of anti-angiogenic cytokines such as IFNs, IL-10, and IL-12 [21].

Among indications for Imiquimod use in dermatologic disorders, at 5% concentration, are basal cell carcinoma, actinic keratosis, and ano-genital warts. In March 2011, the FDA approved Imiquimod 3.75% as a new standard in managing AK affected patients, with the target being the detection and clearance of clinical and subclinical lesions across an entire sun-exposed field.

Patients were evaluated at baseline and at the end of the application of the topical treatment. Lesions were counted and photographed at baseline and at every clinical examination.

### 2.2. Data Acquisition

At the baseline, the number of visible AKs on each patient ranged from 3 to 30, with an average number of 12. One visible lesion was chosen by the expert dermatologists for the NIRS signal analysis, based on (a) lesion position, to facilitate signal acquisition, and (b) lesion evolution, to consider lesions that were most likely to respond to topical treatment. A commercial device (NIRO^®^ 200-NX, Hamamatsu) was used to acquire the NIRS signals at a sampling rate of 5 Hz. The device acquires the change in the relative concentration of the O_2_Hb and HHb and displays them in real time. The device has two probes: an emission probe (LED light source with 735 nm and 810 nm wavelengths) and detection probe (a photodiode). The output power is less than 2 mW. The measurement method was spatially resolved spectroscopy (SRS) and modified Beer–Lambert (MBL). For each patient, the signals were acquired on the following three skin areas: (i) healthy skin (no UVR damage), skin area with AKs (ii) before and (iii) after the treatment. Firstly, we acquired the signals on the healthy skin and the AKs, before any treatment. At the end of the therapy, we acquired the signals on the skin areas that previously showed the lesions and which were cleared, to analyze the regression of the lesions.

The position of the NIRS emission probe and detection probe on the skin influences the depth of the analyzed area and consequently the zone of the body that will be evaluated. For this study, the two probes were placed at a distance of 2.5 cm from each other, so that the path was close to the skin and would not penetrate too deep. In the case of AK, the probes were placed so that the lesion was positioned exactly between the two probes. For the healthy skin area, the probes were positioned on the underside of the arm (while relaxed). After the treatment, the probes were placed on the same area of the AKs acquisition before the therapy (the correct position is displayed in the photos taken before the treatment, Figure 1). The signals were acquired with the same acquisition protocol as in our previous study [20], at least 14 days after stopping the treatment. Briefly, the NIRS signals were acquired over three specific moments: baseline (1.5 min), application of an ice pack near probes as a vasoconstriction stimulus (1.5 min), removal of ice pack and acquisition of vascular recovery (1.5 min).

During the ice pack application, the signals were often corrupted by motion artefacts caused by the initial placement of the ice pack, and for this reason, only the baseline and after epochs were considered in the statistical analysis.

Figure 2 shows an example of AK before (A), after the therapy (B), and the HHb and O_2_Hb signals acquired on the AK before and after treatment (C,D).

### 2.3. Pre-Processing and Feature Extraction

MATLAB R2019b on a Lenovo computer (processor: Intel^®^ Core™ i7-8700 CPU @ 3.20 GHz) was employed for all signal processing and statistical analysis. As a first step before feature extraction, the raw acquired NIRS signals were filtered to exclude artefacts as much as possible.

We applied a 5th order band-pass Chebyshev filter to remove the frequency component outside the range 10–250 mHz. The signals (duration: 4.5 ± 0.5 min) were divided into three different epochs: *before* (before the application of the ice pack), *ice* (during the stimulus of the ice pack), and *after* (after removing the ice pack). The signals were analyzed in the spectral and time domain, and specific features were extracted from each epoch.

The calculated features were 18: 4 in the spectral domain and 14 in the time domain, as listed in Table 1, and briefly explained hereafter. Specifically, the approximate entropy [22] is an indicator of the complexity of the time series, which may be useful to detect several pathological or physiological conditions [23,24,25]. For the computation of the approximate entropy (ApEn), we set
m=2 and r equal to 0.2 times the standard deviation of the signal [22]. The Hjorth parameters are activity, mobility (HM) and complexity (HC) [26]. They are entirely estimated in the time domain, as a function of the variance of the signal and its derivative. However, they also give a frequency domain interpretation, as statistical moments of the power spectrum. The activity is quantified by means of the amplitude variance of the time function and the frequency domain interpretation can be the surface of the power spectrum. Considering the time domain, the activity measures how far the value of the dataset is spread out from the average value and coincides with the variance of the signal. The Hjorth mobility is the square root of the ratio between the variance of the first derivative and the variance of the signal. Since the variance of the first derivative and the variance of the signal are dependent on the mean amplitude, the ratio will be dependent on the curve shape only. In the frequency domain, it represents the mean frequency. The Hjorth complexity is the ratio between the Hjorth mobility of the first derivative of the signal and the Hjorth mobility of the signal. It is dimensionless and, due to the non-linear calculation of standard deviation, this parameter will quantify any deviation from the sine shape. Considering the frequency domain, it represents the change in frequency. The frequency domain features that were extracted were spectral entropy, mean frequency, and normalized power in two bands of interest. The frequency bands of interest were the very low-frequency (VLF = 20 mHz–60 mHz) and low-frequency (LF = 60 mHz–140 mHz) bands. The power in the frequency bands was calculated using a time–frequency transform with a Choi–Williams exponential kernel (sigma = 0.5) [27]. Moreover, the VLF and LF power bands were calculated within each epoch, considering both oxygenated and deoxygenated hemoglobin. The values of VLF and LF bands were normalized considering the total power band of the epoch. The spectral entropy (SE) of a signal is a measure of its spectral power distribution and it is an index of the irregularity and complexity of the signal.

### 2.4. Statistical Analysis

For the statistical analysis, we employed a one-way multivariate analysis of variance (MANOVA). This analysis extends upon the univariate analysis of variance (ANOVA), where statistical differences between two groups considering one continuous variable (i.e., one of the estimated features) are verified. The MANOVA extends this analysis by taking a set of multiple continuous variables and grouping them together into a weighted linear combination. In this way, the canonical variables are estimated, which represent the new features formed by a linear combination of the original features. In our study, the 18 time and frequency variables computed for each patient are the multiple continuous variables, and the two analyzed groups are the same subjects, pre and post topical IMI treatment. After the MANOVA analysis, we further analyzed the features with the highest weights in the first canonical variable. Hence, the MANOVA analysis was applied to evaluate whether there was a difference between the NIRS signals (representing the vascular response) acquired from the lesioned and treated skin areas.

We calculated the features of the lesioned and treated skin and normalized them with respect to the features that were calculated on the healthy skin, in order to evaluate the regression of the lesion with respect to the healthy skin. This step was also done so that the final features used in the statistical analysis on both the *before* and *after* NIRS signals were patient-specific. Hence, the features of each patient were normalized with those of the corresponding healthy skin, considering both the O_2_Hb and HHb signals in the *before* and *after* epochs. The *p*-value and MANOVA dimension (d) were used to determine whether the two groups showed statistically significant differences and could be considered as belonging to two separate groups (d = 1). The statistical analysis was composed of 4 MANOVA tests, one for every epoch (*before* and *after* O_2_Hb and HHb signals). We applied the multiple testing Bonferroni correction [28] in order to demonstrate the statistically significant differences in these multiple tests. Applying the Bonferroni correction, the new significance level was equal to α = 0.05/4 = 0.0125.

## 3. Results

At the baseline, the number of visible AKs on each patient ranged from 3 to 30, with an average number of 12; at the end of the second cycle of topical treatment with Imiquimod 3.75% (IMI), the average number was reduced to 8, and 6 months after the end of treatment, the average number was equal to 3; in particular, in seven patients, a complete clearance of all lesions was achieved (no AKs detectable). Moreover, the specific lesion that was studied with the NIRS analysis was completely cleared in all patients.

NIR spectroscopy was performed, as detailed in the Materials and Methods section, before the treatment and not earlier than 14 days after stopping the second cycle of treatment. The evaluation was not carried out at the end of the first cycle, due to the presence of treatment-related erythema and local inflammatory skin reactions that could change the blood oxygenation status.

In the baseline condition, when the ice pack had not been applied, the epochs of both the HHb and O_2_Hb signals did not present significant differences (*p* > 0.0125). The *after* epochs (i.e., after ice pack application and removal) provide more information about the vascular response, since they were acquired after the cold stimulation and during the vascular recovery. In the HHb signals, the *after* epochs showed a *p*-value < 0.0125 and d = 1, according to the MANOVA analysis. This confirms that the features extracted from the NIRS signals in the different skin areas can be divided into two separate groups (AKs and treated skin), and the first canonical variable is sufficient to discriminate the differences between them (Figure 3).

Through the MANOVA analysis, the coefficients of the new linear combination were estimated and we found the three features for each epoch that had the highest coefficients (in absolute value) for the first canonical variable, which are reported in Table 2, where the bold values are the three highest values. The obtained *p*-values are also listed for each epoch in Table 2, and the epochs denoted with an asterisk show those that presented a *p*-value < 0.0125 (Bonferroni corrected significance level). The Hjorth parameters showed the highest weight in all of the epochs. In addition, the approximate entropy (ApEn) showed a high canonical variable coefficient value in the analysis, especially in the O_2_Hb *after* epoch. The features with the highest weights were always the HC, HM, ApEn, mean, Average Rectified Value (ARV), and median value.

## 4. Discussion

In our previous study [20], we demonstrated that there is a difference in the NIRS features between healthy skin and AK lesions which can be interpreted as a different vascular response in the two areas, using the same protocol and a similar statistical analysis approach. In this study, we advanced the previous findings by evaluating the NIRS signals acquired on the AK lesions and the same skin area after topical treatment with IMI, which can be interpreted as a quantitative evaluation demonstrating the lesion’s response to the treatment. To evaluate the evolution of the lesions and their potential regression, we maintained the relationship between the lesioned skin (before and after treatment) with the healthy skin. In order to do this, we applied the MANOVA, considering the first group as the ratio between the parameters calculated on AK lesions before the treatment and the parameters calculated using the NIRS signals acquired on healthy skin with the same protocol, and the second group as the ratio between the skin area after the treatment and the healthy skin. In this way, the correlation between the AK lesion before and after treatment and the healthy skin of the same patient was maintained. Furthermore, the signals were acquired on the same skin area before and after treatment, in order to evaluate only the effects of the therapy. Indeed, if the signals were acquired on two different skin areas, the different vascular conditions could be caused by several factors (e.g., the different vascular network) and not by only the effects of the topical treatment.

We applied the MANOVA to evaluate whether the two groups showed significant differences, studying both the baseline conditions and the vascular response (after the stimulus). In the baseline condition, the analysis aimed to understand the vascular condition without any external stimulus, to see if there was a different original condition between these two areas. To evaluate the effects of the vasoconstriction stimulus, we analyzed the after epochs, to understand if the vascular response permitted better discrimination between these two groups.

Considering the overall statistical analysis and correcting for the multiple comparisons, none of the *before* epochs showed a statistically significant difference (*p* > 0.0125), whereas after the application of the ice pack, the HHb signals presented a *p*-value smaller than 0.0125. In Figure 3, it can be appreciated how the MANOVA plots of the *after* epochs present a more distinct separation when compared to before the stimulation, also in the case of the O_2_Hb signals, even if not statistically significant. This result explains the importance of the stimulus that improved the discrimination of the different vascular responses before and after topical treatment. The key in this study appears to be the cold stimulation, which induces an initial vasoconstriction and a subsequent vasodilation. It generates a different vascular response both between the lesioned and the healthy skin [20] and between the lesioned and the treated skin.

To date, several studies have evaluated the variations in hemoglobin content in AK in the course of different treatments, but none of them have considered the response to topical IMI treatment. Optical skin imaging obtained by an ANTERA 3D camera [29] has allowed the detection of a global reduction in HHb concentration in the lesion after different therapies (i.e., [30,31]). At the same time, it has confirmed the presence of increased vascularization in AK before treatment [30,31]. The statistically significant differences in the after epochs of the HHb signal found in this study are in accordance with these previous findings. Although, here, the relative concentrations were analyzed, this study confirms a statistically significant difference in the HHb concentration variation after a vasoconstriction stimulus. Conversely, the present study was not focused only on the HHb concentration but on both HHb and O_2_Hb concentration variations as an indicator of vascularization and blood flow. Since we have detected a different response to a vasoconstriction stimulus, we can speculate that the application of topical IMI may act also at a vascular level. Accordingly, a few recent studies have highlighted an antiangiogenic effect of topical IMI, thus suggesting a possible use in the treatment of hemangiomas [32,33].

Applying the MANOVA, we found that the first canonical variable is enough to discriminate between the two groups. Further analyzing the features with the highest weights in the first canonical variable, we found that the features were always time domain parameters: the Hjorth parameters, approximate entropy, the mean/median value, and the ARV. An innovative result was the influence and importance of the Hjorth parameters, which are usually used to analyze the EEG signals [26], both in the *before* and *after* epochs. These parameters are entirely based on a time domain analysis, but they can be derived also from the statistical moments of the power spectrum [34]. The Hjorth complexity (HC) was among the three features with the highest weight in all four considered cases, whereas the Hjorth mobility (HM) was among the three features with the highest weight in all cases except for O_2_Hb *before* epoch. A signal with a higher value of variance shows a higher value of HM, and a signal that shows a greater change in frequency in the epoch will demonstrate higher values of HC. The ApEn, an index of the signal complexity, showed the highest weight in the HHb *after* epochs. The median values, the mean value, and ARV show that the concentration and saturation of the hemoglobin changed differently. Examining the specific signals more closely, considering the concentration of the O_2_Hb, different variations and oscillations are shown in the O_2_Hb signal before and after the therapy, which is confirmed by the fact that the HM and ApEn are between the three features with the highest weights. Considering the HHb signal, the Hjorth parameters and median value have a bigger impact in the canonical variable calculation, which is an index of the different vascular networks in the skin area before and after the therapy, that is reflected in a different amount of HHb.

Hence, these parameters were found to yield important information about the vascular condition, representing the different characteristics of the signals in the AK lesions, before and after treatment. The different levels of complexity of the HHb and O_2_Hb relative concentrations during the vascular response after the cold stimulus could be explained by the abnormal vascular network that is typically shown in pathological tissues that generates a different vascular condition and different vasodilation. In fact, studies have shown how histological changes common to photodamaged skin, such as AKs, include increased vascularity and dilated vessels, among other things [2]. This increased vascularity and dilated vessels that are present in AKs influence the blood flow and the blood vessel response to the cold stimulation—in particular, in the complexity of the hemoglobin concentration variations. In fact, since the non-invasive NIRS analysis provides relative, and not absolute, changes in oxygenated and deoxygenated hemoglobin, this difference in vasculature of the lesion is not evidenced during the baseline *before* epoch (*p* > 0.0125). On the other hand, after a vasoconstrictive cold stimulation is applied to the skin, the NIRS signals are able to capture this difference in vasculature between lesioned and treated skin, which is particularly noticeable in the HHb signals.

In our previous study [20], we discovered the importance of the Hjorth parameters for the discrimination between the AK lesions and the healthy skin. Interestingly, the spectral parameters did not show higher weights, although the time and spectral parameters are correlated. In the O_2_Hb *after* epochs, the ApEn showed a high canonical variable coefficient in our previous study, and this finding was also confirmed in this study. Hence, the time domain parameters were shown to be important in both comparisons: (1) between the healthy and lesioned skin [20] and (2) in the lesioned skin before and after treatment, which is presented in this study.

While the overall results of this study are encouraging, there are some limitations. First of all, the size of the considered database is limited. It is necessary to include more patients and cases in order to confirm these initial findings. Secondly, the diagnosis of the lesion and the subsequent effect of the treatment was analyzed only visually, and we have no histological results to confirm the NIRS analysis results and interpretation. Moreover, with the expansion of the dataset and with the inclusion of histological data and other clinical data, a sophisticated machine learning study to classify the skin area using feature selection or a feature importance analysis can be employed to help answer some important questions, such as how accurately we can use the parameters computed on the NIRS data. Thirdly, while the acquisition protocol is easy to perform, there are some limitations to its standardization. Specifically, while the orientation and alignment of the NIRS probes should not substantially affect the acquired NIRS signals as long as they cover the same lesion area, we did not specifically test the probes’ position robustness. Furthermore, the ice pack application is an important step in the acquisition protocol but requires further studies to standardize the optimal application location with respect to the probes and lesion.

## 5. Conclusions

In this study, we demonstrate that the NIRS and cold stimulation approach has clear potential in the non-invasive diagnosis of AK lesions and in the quantitative analysis of their regression after topical treatment. Through this non-invasive tool, we demonstrate that the AK lesions showed a statistically different response to the topical treatment, which could assist in the diagnosis of AK lesions and in the evaluation of their regression after treatment.

The technique can be used alongside the traditional clinical and histopathological examination and could help to classify healthy skin and AK lesions, as well as to monitor the capability of a specific treatment to drive the lesion towards regression. Furthermore, NIR spectroscopy is not painful for patients and the acquisition protocol is easy, repeatable, and practical for clinicians. We demonstrated that a non-invasive vasoconstriction stimulus (i.e., application of the ice pack) evidences the differences between the AK lesions and the treated skin, confirming the effects of the topical treatment. We are currently planning on enlarging the study to include more patients and to include also histological analysis and results to confirm the ability of this non-invasive technique to discriminate between healthy and lesioned skin and to progressively evaluate the evolution of a lesion or the effects of a treatment. In the future, we will also attempt to apply such technology in other kinds of preneoplastic and neoplastic cutaneous lesions, thus including hyperkeratotic AK, Bowen disease, and invasive SCC.

## Figures and Tables

**Figure 1 medicina-56-00482-f001:**
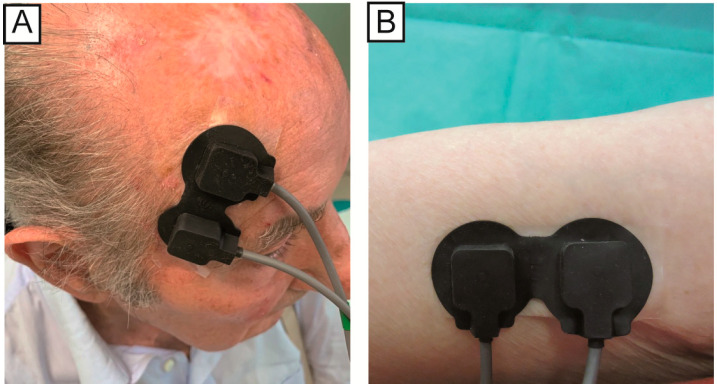
The position of the probes in the actinic keratosis (AK) lesions (**A**) and healthy skin (**B**).

**Figure 2 medicina-56-00482-f002:**
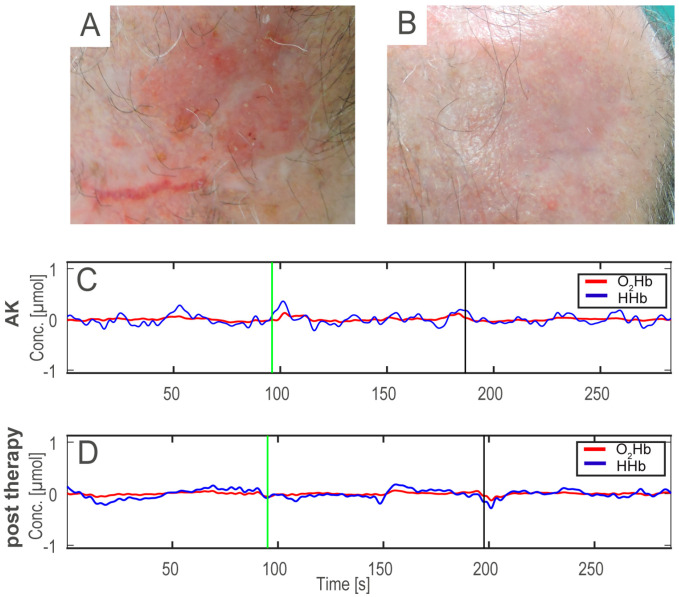
First row: example of AK before (**A**) and after the treatment (**B**), showing the typical AK features with erythema, telangiectasias, and slight hyperkeratotic scales (**A**), that were both strongly reduced after therapy (**B**). Second and last row: example of oxy-hemoglobin and deoxy-hemoglobin (O_2_Hb and HHb) signals acquired on the AK lesion before (**C**) and after the treatment (**D**). In panels (**C**,**D**), the HHb signals are blue and the O_2_Hb signals are red. Conc: micro moral (µMol) concentration.

**Figure 3 medicina-56-00482-f003:**
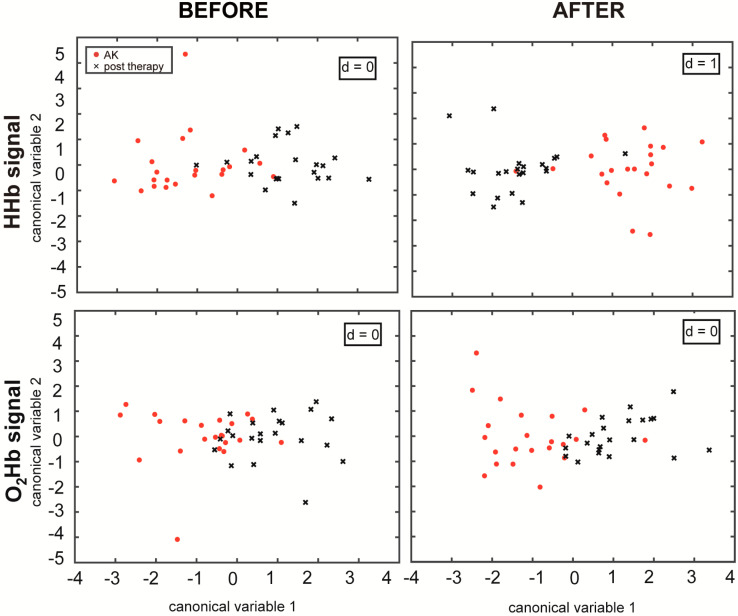
The first two canonical variables of the features of the HHb and O_2_Hb signals as calculated by MANOVA.

**Table 1 medicina-56-00482-t001:** Mathematical description of features.

Feature Name	Mathematical Description
Mean	$\bar{x}=\frac{1}{N}\overset{N}{\underset{i=1}{\sum }}x_{I}$
Variance	$var\left(x\right)=\frac{1}{N}\overset{N}{\underset{i=1}{\sum }}{\left(x_{i}-\bar{x}\right)}^{2}$
Median	Middle value of order numbers
Maximum	The maximum value of time series
Peak-to-Peak Amplitude	Difference between the maximum positive and negative amplitudes
Skewness	$skew\left(x\right)=\frac{\frac{1}{N}\sum_{i=1}^{N}{\left(x_{i}-\bar{x}\right)}^{3}}{{\left(\sqrt{var\left(x\right)}\right)}^{3}}$
Kurtosis	$Kurt\left(x\right)=\frac{\frac{1}{N}\sum_{i=1}^{N}{\left(x_{i}-\bar{x}\right)}^{4}}{{\left(var\left(x\right)\right)}^{2}}$
Hjorth Mobility	$HM\left(x\right)=\sqrt{\frac{var\left(\frac{dx}{dt}\right)}{var\left(x\right)}}$
Hjorth Complexity	$HC\left(x\right)=\sqrt{\frac{HM\left(\frac{dx}{dt}\right)}{HM\left(x\right)}}$
Approximate Entropy	$\mathrm{ApEn}=\frac{1}{N-\left(m-1\right)}\overset{N-\left(m-1\right)}{\underset{i=1}{\sum }}\ln C_{i}^{m}\left(r\right)-\frac{1}{\left(N-m\right)}\overset{N-m}{\underset{i=1}{\sum }}\ln C_{i}^{m+1}(r)$
Average Rectified Value (ARV)	$ARV\left(x\right)=\frac{1}{N}\overset{N}{\underset{i=1}{\sum }}\left|x_{i}\right|$
Root mean square RMS	$RMS\left(x\right)=\sqrt{\frac{1}{N}\overset{N}{\underset{i=1}{\sum }}{\left|x_{i}\right|}^{2}}$
Energy	$E\left(x\right)=\frac{1}{N}\overset{N}{\underset{i=1}{\sum }}{\left|x_{i}\right|}^{2}$
Number of Zero Crossing	Number of times the signal crosses *x*-axis
Power VLF	P_VLF_, the spectral power between 20 and 60 mHz
Power LF	P_LF_, the spectral power between 60 and 140 mHz
Spectral Entropy	$SE=\overset{N}{\underset{i=1}{\sum }}p_{i}\cdot \ln\frac{1}{p_{i}}$
Mean Frequency	$MNF=\frac{\sum_{i=1}^{N}P_{i}\cdot f_{i}}{\sum_{i=1}^{N}P_{i}}$

*P* is power spectrum; *p* is normalized power spectrum; $C_{i}^{m}\left(r\right)$ is the fraction of pattern of length m; r is the criterion of similarity. Power VLF is the very low frequency spectral power (range: 20–60 mHz) and Power LF is the low frequency spectral power (range: 60–140 mHz)

**Table 2 medicina-56-00482-t002:** All features and their canonical variable coefficients as calculated by MANOVA. For each analyzed epoch, the MANOVA *p*-value is listed. The bold values are the three highest values

Features	Coefficients HHb before	Coefficients O_2_Hb before	Coefficients HHb after *	Coefficients O_2_Hb after
	*p*-Value 0.025	*p*-Value 0.398	*p*-Value 0.006	*p-*Value 0.084
Mean	0.60	**5.00**	0.86	2.61
Variance	0.01	0.14	−0.06	0.31
Maximum Value	0.14	0.28	−0.01	−0.29
Median	**4.31**	0.50	**3.55**	−7.02
Peak-to-Peak Amplitude	−0.04	−0.05	0.01	0.12
Kurtosis	0.36	0.06	0.22	−0.10
Skewness	0.39	−0.06	0.79	−0.64
Spectral Entropy	0.28	1.90	−0.98	−0.92
Approximate Entropy	−3.76	−2.68	1.66	**−11.06**
RMS	−0.54	0.92	−0.93	0.44
Average Rectified Value (ARV)	−2.33	**−3.25**	1.36	−0.88
Hjorth Mobility	**14.34**	0.68	**17.96**	**46.44**
Hjorth Complexity	**11.97**	**18.85**	**8.98**	**35.14**
Number of Zero Crossing	−0.21	−0.06	0.18	−0.12
Energy	0.00	0.00	0.00	0.00
Power VLF	0.11	0.13	−0.03	0.18
Power LF	−0.35	0.61	0.13	−0.03
Mean Frequency	−0.23	−0.48	−0.13	−1.14

* denotes a statistically significant difference with the Bonferroni correction.

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
