# Peer review of "Non-Invasive Analysis of Actinic Keratosis before and after Topical Treatment Using a Cold Stimulation and Near-Infrared Spectroscopy"

_medicina, 2020, doi:10.3390/medicina56090482_

Round 1

Reviewer 1 Report

This is original and interesting study adressing an unmet medical need concerning the lacking of procedure able to monitor therapeutic results of treatments mirated to AKs.

The described procedure based on the employment of near-infrared spectroscopy, seems to give future opportunity in overcoming of the actual limits in interpreting therapeutic results of topical treatments for AK.

The translational research message of this study could be significant in clinical setting.

English lenguage should be somewhat improved.

Author Response

Dear Editor,

We thank you and the reviewers for the interest and comments on our paper entitled “Non-invasive analysis of actinic keratosis before and after topical treatment using a cold stimulation and near-infrared spectroscopy”.

In the present version of the manuscript we tried to implement the reviewer suggestions.

In detail, according the reviewer’s suggestion, we performed the following changes:

Reviewer 

This is original and interesting study adressing an unmet medical need concerning the lacking of procedure able to monitor therapeutic results of treatments mirated to AKs.

The described procedure based on the employment of near-infrared spectroscopy, seems to give future opportunity in overcoming of the actual limits in interpreting therapeutic results of topical treatments for AK.

The translational research message of this study could be significant in clinical setting.

English lenguage should be somewhat improved.

We thank the reviewer very much for the positive interest on our work and for understanding its translational message.

According to his suggestion, the English language has been revised and improved.

Reviewer 2 Report

RThe study describes a novel assessment tool for actinic keratosis using cold stimulation and near-infrared spectroscopy. 

A small number involving 22 males was used to study the efficacy of the assessment tool. The early results are promising and a larger evaluation is clearly needed.

The main concern with this manuscript is the background material, which acts as the justification for the assessment tool.

Actinic keratoses (AKs) are regarded as a form of SCCis. Whilst aome accept this understanding, many do not.

Similarly, the term field cancerisation implies that the actinic  damage is a low grade cancer.

Further the authors presume that managing field actinic damage will reduce future skin cancer risk. We know that various field treatments reduce the observed actinic damage in a skin field. We also now that a small percentage of AKs will become invasive SCC in time. What is not yet clear is whether field treatment reduces subsequent SCC incidence in the field treatment. 

The references cited are concerning. Most are dated and many major studies are absent. 

The authors may wish to avoid suggesting that AKs are a low grade SCCis. The authors may prefer to use "field actinic damage" rather than "field cancerization". Such changes would leave the controversy out of the background and hance allow a greater focud on the study proper.

Regarding the conclusions, it remains unclear why this assessment tool should be prefered over traditional clinical assessment and digitial photography follow up of AKs before and after field tretament. 

The authors quote Stockfleth (2017). The imprtance of treating the field in actnic keratosis. This paper concludes that, "Additionally, preclinical studies suggest that field therapy may prevent or delay the recurrence of non-melanoma skin cancer.". Indeed, all we know is that future skin cancers may be altered by field treatment. Some studies have shown that field treatment has not altered subsequent cancer burden, even though the actinic damage was visibly improved. As such, the importance of treating field damage is currently small, largely cosmetic, as the future cancer burden possible benefit is unknown.

Author Response

Dear Editor,

We thank you and the reviewers for the interest and comments on our paper entitled “Non-invasive analysis of actinic keratosis before and after topical treatment using a cold stimulation and near-infrared spectroscopy”.

In the present version of the manuscript we tried to implement the reviewer suggestions.

In detail, according the reviewer’s suggestion, we performed the following changes:

The study describes a novel assessment tool for actinic keratosis using cold stimulation and near-infrared spectroscopy. 

A small number involving 22 males was used to study the efficacy of the assessment tool. The early results are promising and a larger evaluation is clearly needed.

We are aware of the relatively low number of the samples examined and of the need to expand the study on a greater number of cases. However, in consideration of the novelty of the diagnostic approach, we think that also these preliminary results may be of interest.

The main concern with this manuscript is the background material, which acts as the justification for the assessment tool.

Actinic keratoses (AKs) are regarded as a form of SCCis. Whilst aome accept this understanding, many do not.

Similarly, the term field cancerisation implies that the actinic  damage is a low grade cancer.

We believe that actinic keratoses should be considered carcinomas in situ with a non inconsistent risk of transformation into SCC. However, we are aware of the uniqueness of view present in the literature and we have modified the abstract and introduction in accordance with the reviewer's suggestion (lines 13-15, 34-39, 41-47, 386-388, 395-398).

The possible evolution of actinic keratoses (AKs) into invasive squamous cell carcinomas (SCC) makes their treatment and monitoring essential.

Non-melanoma skin cancers (NMSC) are the most common human cancers, particularly in the Caucasian population; in Italy, the prevalence of the Actinic Keratoses (AKs) is estimated to be about 30%[1], making them a major health problem, in consideration of their possibility of evolution into invasive squamous cell carcinomas (SCC).

It is well-known that the chronic exposure to ultraviolet radiation (UVR) is the main risk factor for NMSC and UVR-induced photo damage can cause a field of actinic damage. This field usually presents clinically visible AKs and subclinical cell alterations (histologic atypia), that represent a substrate for new lesions development [2–5]. The early treatment of all lesions (clinically visible and subclinical) must be considered to reduce the possible risk of evolution into Squamous Cell Carcinoma (SCC) [6-11].

The technique can be used alongside the traditional clinical and histopathological examination and could help to classify healthy skin and AK lesions, and also to monitor the capability of a specific treatment to drive the lesion towards regression.

In the future, we will also attempt to apply such technology in other kinds of preneoplastic and neoplastic cutaneous lesions, thus including hyperkeratotic AK, Bowen disease and invasive SCC.

Further the authors presume that managing field actinic damage will reduce future skin cancer risk. We know that various field treatments reduce the observed actinic damage in a skin field. We also now that a small percentage of AKs will become invasive SCC in time. What is not yet clear is whether field treatment reduces subsequent SCC incidence in the field treatment. 

The references cited are concerning. Most are dated and many major studies are absent. 

The authors may wish to avoid suggesting that AKs are a low grade SCCis. The authors may prefer to use "field actinic damage" rather than "field cancerization". Such changes would leave the controversy out of the background and hance allow a greater focud on the study proper.

We modified some paragraphs of the introduction in accordance with the reviewer suggestions and added appropriate new references. However, we would like to underline that the purpose of our study is not to demonstrate the possible evolution of AKs in SCCs, but to describe an innovative tool for the clinical monitoring of AKs. We therefore added the following comment in the introduction (lines 34-39, 41-47):

Non-melanoma skin cancers (NMSC) are the most common human cancers, particularly in the Caucasian population; in Italy, the prevalence of the Actinic Keratoses (AKs) is estimated to be about 30%[1], making them a major health problem, in consideration of their possibility of evolution into invasive squamous cell carcinomas (SCC).

It is well-known that the chronic exposure to ultraviolet radiation (UVR) is the main risk factor for NMSC and UVR-induced photo damage can cause a field of actinic damage. This field usually presents clinically visible AKs and subclinical cell alterations (histologic atypia), that represent a substrate for new lesions development [2–5]. The early treatment of all lesions (clinically visible and subclinical) must be considered to reduce the possible risk of evolution into Squamous Cell Carcinoma (SCC) [6-11].

Regarding the conclusions, it remains unclear why this assessment tool should be prefered over traditional clinical assessment and digitial photography follow up of AKs before and after field tretament. 

We think that the NIRS signal analysis technique can give additional information to the traditional clinical assessment, but not replace the clinical examination. This was better explained in the conclusion (lines 386-388, 395-398).

The technique can be used alongside the traditional clinical and histopathological examination and could help to classify healthy skin and AK lesions, and also to monitor the capability of a specific treatment to drive the lesion towards regression.

 In the future, we will also attempt to apply such technology in other kinds of preneoplastic and neoplastic cutaneous lesions, thus including hyperkeratotic AK, Bowen disease and invasive SCC.

The authors quote Stockfleth (2017). The imprtance of treating the field in actnic keratosis. This paper concludes that, "Additionally, preclinical studies suggest that field therapy may prevent or delay the recurrence of non-melanoma skin cancer.". Indeed, all we know is that future skin cancers may be altered by field treatment. Some studies have shown that field treatment has not altered subsequent cancer burden, even though the actinic damage was visibly improved. As such, the importance of treating field damage is currently small, largely cosmetic, as the future cancer burden possible benefit is unknown.

As previously reported, the main goal of our work is not to discuss the relevance of the treatment of the cancerization field. Instead, we propose a new tool to monitor these lesions.

We hope that, in the present form, our paper will be suitable for publication in the Special Issue of ”Medicina.

Round 2

Reviewer 2 Report

The previous concerns about ussge of terms like "field cancerization" have been appropriately addressed. 

This manuscript is a resubmission of an earlier submission. The following is a list of the peer review reports and author responses from that submission.

Round 1

Reviewer 1 Report

The aim of our study is to determine if a time and frequency analysis of the NIRS (near infrared spectroscopy) signals acquired on the skin lesion before and after a topical treatment can provide quantitative differences between the AK skin lesion area before and after a topical treatment. This is an interesting observation using NIRS to access the lesions of AK noninvasively. The comments are listed below. (1) Would the orientation and alignment of the two NIRS probes affect the signals of AK in between? It might be important to incorporate this factor into the validation and consistency of NIRS signals from AK. (2) Ice packing may be difficult to standardize. Would it be possible to combine the NIRS probes with an iontophoretic device that delivers sympathomimetics? (3) The incorporation of the regular optic photos of AK might be included. Since the telangiectasis may be an important feature of AK. How the vascular blood flow might be different in AK and in normal skin is interesting to know. The summation of HHb and O2Hb may be an indicator of total skin blood flow. (4) The assessment of the normal skin is important to this study. It seems like that the authors are comparing the signals before and after treatments for AK. However, normal skin is not measured. (5) Editorial errors, for example, Line 363, “In” is repeated twice.